# Proposal of a New Prognostic Model for Differentiated Thyroid Cancer with *TERT* Promoter Mutations

**DOI:** 10.3390/cancers13122943

**Published:** 2021-06-11

**Authors:** Jun Park, Sungjoo Lee, Jiyun Park, Hyunju Park, Chang-Seok Ki, Young-Lyun Oh, Jung-Hee Shin, Jee-Soo Kim, Sun-Wook Kim, Jae-Hoon Chung, Kyunga Kim, Tae-Hyuk Kim

**Affiliations:** 1Thyroid Center, Samsung Medical Center, Division of Endocrinology and Metabolism, Department of Medicine, Sungkyunkwan University School of Medicine, Seoul 06351, Korea; pjun113@gmail.com (J.P.); nove1123.park@samsung.com (J.P.); hj1006.park@samsung.com (H.P.); swkimmd@skku.edu (S.-W.K.); thyroid@skku.edu (J.-H.C.); 2Division of Endocrinology, Department of Medicine, Sahmyook Medical Center, Seoul 02500, Korea; 3Department of Digital Health, Samsung Advanced Institute for Health Sciences & Technology, Sungkyunkwan University School of Medicine, Seoul 06351, Korea; ldh1223@g.skku.edu; 4Green Cross Genome, Yongin 16924, Korea; cski@gccorp.com; 5Samsung Medical Center, Department of Pathology, Sungkyunkwan University School of Medicine, Seoul 06351, Korea; yl.oh@samsung.com; 6Samsung Medical Center, Department of Radiology, Sungkyunkwan University School of Medicine, Seoul 06351, Korea; helena35.shin@samsung.com; 7Samsung Medical Center, Department of Surgery, Sungkyunkwan University School of Medicine, Seoul 06351, Korea; js0507.kim@samsung.com; 8Statistics and Data Center, Samsung Medical Center, Research Institute for Future Medicine, Seoul 06355, Korea

**Keywords:** differentiated thyroid cancer, *TERT* promoter, TNM staging, prognosis, mortality

## Abstract

**Simple Summary:**

In a cohort study involving 393 patients with differentiated thyroid cancer, we incorporated the *TERT* mutational status into AJCC TNM staging 8th edition (TNM-8), proposing a new prognostic system, termed TNM-8T. We demonstrated that the new prognostic system is superior to the existing TNM-8 staging system and its contents are not very complicated. This study is the first to present a new model that combines the mutational profile with the AJCC TNM stage to improve the predictability of cancer-specific survival with long-term follow-up.

**Abstract:**

The role of telomerase reverse transcriptase (*TERT*) promoter mutations as an independent poor prognostic factor in differentiated thyroid cancer (DTC) patients is well known, but there is no prognostic system that combines the *TERT* promoter mutation status with tumor-node-metastasis (TNM) stage to predict cancer-specific survival (CSS). A total of 393 patients with pathologically confirmed DTC after thyroidectomy were enrolled. After incorporating wild-type *TERT* and mutant *TERT* with stages I, II, and III/IV of the AJCC TNM system 8th edition (TNM-8), we generated six combinations and calculated 10-year and 15-year CSS and adjusted hazard ratios (HRs) for cancer-related death using Cox regression. Then, a new mortality prediction model termed TNM-8T was derived based on the CSS and HR of each combination in the four groups. Of the 393 patients, there were 27 (6.9%) thyroid cancer-related deaths during a median follow-up of 14 years. Patients with a more advanced stage had a lower survival rate (10-year CSS for TNM-8T stage 1, 2, 3, and 4: 98.7%, 93.5%, 77.3%, and 63.0%, respectively; *p* < 0.001). TNM-8T showed a better spread of CSS (*p* < 0.001) than TNM-8 (*p* = 0.002) in the adjusted survival curves. The C-index for mortality risk predictability was 0.880 (95% CI, 0.665–0.957) in TNM-8T and 0.827 (95% CI, 0.622–0.930) in TNM-8 (*p* < 0.001). TNM-8T, a new prognostic system that incorporates the *TERT* mutational status into TNM-8, showed superior predictability to TNM-8 in the long-term survival of DTC patients.

## 1. Introduction

Telomerase reverse transcriptase (*TERT*) promoter mutations have been detected in several human cancers, including thyroid cancer [1], and have been reported to be an independent poor prognostic factor in the recurrence and cancer-specific survival (CSS) of patients with differentiated thyroid cancer (DTC) [2,3].

The American Joint Committee on Cancer/Union for International Cancer Control (AJCC/UICC) tumor-node-metastasis (TNM) staging system is most commonly used to predict the prognosis of thyroid cancer. To predict disease mortality, AJCC/UICC staging is recommended for all DTC patients according to the 2015 American Thyroid Association guidelines [4]. It was revised to the 8th edition (TNM-8) in 2016 [5,6], improving its predictability of CSS compared to the 7th edition in patients with DTC [7,8]. However, there are some problems. Approximately 30% of DTC patients are downstaged to stage I or II, so there are too few patients with stage III and IV disease [6,9], and there is no molecular profile of clinical prognostic significance that has recently emerged [10,11]. In addition, a large multicenter cohort study showed the poor predictability of TNM-8 in the CSS of follicular thyroid cancer (FTC) patients [12]. Thus, there has been a need for a more complementary staging system in predicting the survival of individual DTC patients.

Kim et al. [13] proposed a new prognostic model by incorporating *TERT* promoter mutations into a dynamic risk stratification (DRS) system and demonstrated that it is as effective as DRS at predicting structural recurrence and CSS in nonmetastatic DTC patients. In addition, our research team recently reported that the prognostic predictability increased by the inclusion of *TERT* mutations based on the TNM-8 system regardless of the histologic type or initial stage of DTC patients [14].

Therefore, in this study, we aimed to develop a new prediction model for CSS and to compare it with the prediction power of the existing TNM-8 staging system with the hypothesis that if the *TERT* promoter mutational status is incorporated into TNM-8, a better prediction system will be created.

## 2. Materials and Methods

### 2.1. Patients and Clinicopathological Data

The same dataset was used in the previous study [14]. In brief, from October 1994 to December 2004, a total of 393 patients with pathologically confirmed DTC, 327 with papillary thyroid cancer (PTC) and 66 with FTC, including Hurthle cell carcinoma, were enrolled in this study after thyroidectomy and neck dissection. All patients received thyrotropin suppression therapy, and 364 (310 with PTC and 54 with FTC) received radioiodine ablation after surgery according to standard guidelines [15,16].

CSS was defined as the time of the initial surgical treatment to the last observation date (31 December 2018) or the date of thyroid cancer-specific death. Patients who died from other causes were censored at the time of death.

One sample per patient was collected from thyroid cancer tissue through the Department of Pathology at Samsung Medical Center, and mutation analysis was performed. Since it was performed after surgery and radioiodine treatment, the result did not affect the decision-making process of the physicians. Thyroid cancer-related mortality data were derived from the Korea National Statistical Office and hospital medical records.

### 2.2. Detection of TERT Promoter Mutations in Thyroid Cancer Samples

Genomic DNA was extracted from formalin-fixed paraffin embedded (FFPE) tissue using the Qiagen DNA FFPE Tissue Kit (Qiagen, Hilden, Germany) according to the manufacturer’s instructions. For FFPE tissue, 4 μm thick unstained slides were prepared, and pathologists decided to use the slides for DNA extraction based on a minimum tumor percentage of 75%. Then, we used seminested polymerase chain reaction (PCR) to identify *TERT* promoter mutations, and mutations were enriched with 3′-modified oligonucleotide PCR.

### 2.3. Statistical Analysis

We stratified DTC patients by a combination of the *TERT* promoter mutation status [wild-type (WT) or mutant] and TNM-8 stage. At this time, as the number of patients in stages III and IV was very small, they were grouped into one group, resulting in a total of six combinations. To propose alternative prognostic groupings, Cox regression was used to calculate the unadjusted and adjusted hazard ratios (HRs) by univariate and multivariate analyses (with Firth’s penalized Cox regression), and 10-year and 15-year survival rates were obtained. Among the variables with a *p* value ≤ 0.2 in the univariate analysis, sex and histology were selected as covariates for the multivariate analysis, except for the age, extrathyroidal extension, distant metastasis, and tumor size constituting TNM-8 (Appendix A). After all pairwise analyses for each of the 6 combinations of TNM-8 stage and *TERT* promoter mutation status, four alternative groups were derived, which were termed TNM-8T: TNM-8 + *TERT*. These four TNM-8T groups were also subjected to pairwise multivariate analysis. Proportional hazards assumption checking was performed with Schoenfeld’s residual test.

Thyroid CSS was analyzed by Kaplan–Meier survival curves and compared with the log-rank test and adjusted survival curves (adjusted for age and histology via a Cox proportional hazards model).

TNM-8T was evaluated through internal validation based on 1000 datasets generated by the stratified bootstrapping technique. The bootstrap sample was used as the training set, and the out-of-bag data for each sampling were used as the testing set. Harrell’s C-index and the integrated area under the curve (iAUC) during the 10-year follow-up were calculated to evaluate the discrimination ability. [17,18] Additionally, the proportion of variance explained (PVE) values and Brier scores at the 10-year follow-up were calculated to evaluate the overall performance of the two staging systems. [19,20] Differences in the C-index, iAUC, PVE, and Brier score between the two staging systems, TNM-8T and TNM-8, were analyzed by paired t-tests.

Statistical analyses were performed using R 3.6.1 (Vienna, Austria; http://www.R-project.org; accessed on 9 November 2020). A *p*-value ≤ 0.05 was considered statistically significant.

## 3. Results

### 3.1. Patient Characteristics

The clinical and genetic characteristics of DTC patients are summarized in Table 1. Of all patients, 329 (83.7%) were female, and 64 (16.3%) were male. The median age at diagnosis was 42.8 years (range 15.8–81.4 years), and 319 (81.2%) were under 55 years of age. Most of the tumors were unifocal (72.8%) and PTC (83.2%), and most patients were classified as TNM-8 stage I (83.7%) at the time of diagnosis and received postoperative radioactive iodine therapy (92.6). *TERT* promoter mutations were identified in 10.9% (43/393, 4 of C250T and 39 of C228T) of patients. As the stage increased, the higher the prevalence of *TERT* mutations was identified (4.9% in stage I, 35.4% in stage II, and 62.5% in stage III/IV). During the median follow-up of 16 years (interquartile range 14–19 years), there were 27 thyroid cancer-related deaths (6.9%).

### 3.2. Alternative Grouping According to the AJCC Stage and TERT Promoter Mutation Status

We analyzed the HRs of six combinations of TNM-8 stage and *TERT* promoter mutation status after adjusting for sex and histologic type for thyroid cancer-related death with 10-year and 15-year CSS (Table 2 and Appendix A). When patients with the same *TERT* mutational status were analyzed, the 10-year and 15-year CSS rates decreased as the stage increased, and the mutant *TERT* groups were associated with lower 10-year and 15-year CSS rates than the WT *TERT* groups regardless of stage (*p* < 0.001) (Table 2). Thyroid CSS was visualized via Kaplan–Meier curves and adjusted survival curves (Appendix A). Based on the above results, an alternative staging system of four groups termed TNM-8T was generated to predict thyroid CSS: TNM-8T stage 1 (patients with TNM-8 stage I and WT *TERT*), TNM-8T stage 2 (patients with TNM-8 stage II and WT *TERT*), TNM-8T stage 3 (patients with TNM-8 stage III/IV and WT *TERT* or with TNM-8 stage I and mutant *TERT*), and TNM-8T stage 4 (patients with TNM-8 stage II or III/IV and mutant *TERT*). The 10-year CSS rates were 98.7%, 93.5%, 77.3%, and 63.0%, respectively (*p* < 0.001) (Table 3 and Table 4). Overall, there was no severe violation of the proportional hazards assumptions of either staging system.

### 3.3. Comparison of Survival in Patients Staged According to TNM-8 and TNM-8T

For comparison, the HRs of each stage of TNM-8T and TNM-8 for cancer-related death after adjusting for sex and histologic type were calculated (Table 4 and Appendix A). In TNM-8T, the adjusted HRs increased as the stage increased (adjusted HR 10.62, 18.57, and 62.87 for stages 2, 3, and 4, respectively), but in TNM-8, there was no difference in the adjusted HRs of stages III and IV (adjusted HRs 13.20, 26.99, and 26.39 for stages II, III, and IV, respectively) (Table 4). TNM-8T showed a better spread of CSS (*p* < 0.001) than TNM-8 (*p* = 0.002) in the adjusted survival curves (Figure 1). The C-index indicating risk predictability was 0.880 (95% CI, 0.665–0.957) in TNM-8T, which was significantly higher than that in TNM-8 (0.827; 95% CI, 0.622–0.930) (*p* < 0.001). The PVE, which indicates the ability of the two staging systems to explain CSS, was 0.205 (95% CI, 0.151–0.264) in TNM-8T, which was also significantly higher than that in TNM-8 (0.153; 95% CI, 0.095–0.221) (*p* < 0.001).

### 3.4. Comparison of TNM Staging Groups According to TNM-8 vs. TNM-8T

Of the 393 DTC patients, the stages of 42 (10.7%) were changed in TNM-8T. Of the 329 patients with stage I disease, 16 were upstaged to stage 3, and of the 48 patients with stage II disease, 17 were upstaged to stage 4. Six patients with stage III disease were upstaged to stage 4, and 3 patients with stage IV disease were downstaged to stage 3. Figure 2 demonstrates the stage migration in our data from TNM-8 to TNM-8T based on the alluvial flow diagram.

## 4. Discussion

DTC is generally an indolent tumor with low mortality, but in some patients, it often progresses aggressively, and distinguishing high-risk patients early in diagnosis is very important for providing the best clinical outcomes for individual patients.

The importance of genetic markers in predicting the prognosis of thyroid cancer has already been reported [10,21]. Among them, the prognostically most promising and notable are carcinogenic mutations and *BRAF* V600E and *TERT* promoter mutations. The poor outcomes caused by *BRAF* V600E and *TERT* promoter mutations or their synergistic effects have been well characterized and widely appreciated [22,23,24,25,26].

The *TERT* promoter has two mutational hotspots on chromosome 5, C228T and C250T. *TERT* C228T is far more prevalent than C250T in thyroid cancer. Both mutations play carcinogenic roles by generating a new site where E-twenty-six transcription factors can bind and increase the transcriptional activities of the *TERT* promoter [27]. Those with less differentiated histologic types have a high prevalence (on average, 11.3% of PTCs, 17.1% of FTCs, 43.2% of poorly differentiated DTCs, and 40.1% of anaplastic thyroid cancers) [22,24]. They are also associated with aggressive tumor behaviors, radioiodine refractoriness, and increased tumor recurrence and DTC-specific death [2,28,29,30,31]. In particular, in PTC patients, coexisting *BRAF* V600E and *TERT* promoter mutations are associated with increased cancer aggressiveness, lymph node and distant metastases, and an advanced stage and increase tumor recurrence and mortality more rapidly than either alone [32,33,34]. This study also showed consistent results with those of previous studies, in which all patients with mutant *TERT*, regardless of stage, had lower 10-year and 15-year CSS rates and higher HRs than those with WT *TERT* when the six combinations of TNM-8 stages and *TERT* mutational statuses were assessed (Table 2).

PTC accounts for 90% of all DTCs [35,36], and *BRAF* mutations are found in approximately 45% of sporadic PTC tumors [37]. In particular, more than 80% of patients newly diagnosed with PTC in Korea have *BRAF* mutations [38]. Therefore, it is believed that *TERT* promoter mutations play an important role in death from DTC. In a previous study, we demonstrated that *TERT* promoter mutations act as an independent poor prognostic factor with the *TERT* promoter mutational status based on TNM-8 in a cohort of patients with DTC [14].

Therefore, in this study, we proposed a new prognostic staging system, TNM-8T, by incorporating the *TERT* promoter mutational status into the AJCC staging system, the most commonly recommended staging system for predicting the prognosis of thyroid cancer. The adjusted HRs significantly increased in all pairwise multivariate Cox regression analyses (*p* < 0.001) (Appendix A). In addition, through various internal validations, it was demonstrated that the TNM-8T staging system has a C-index of 0.880 (95% CI, 0.665–0.957), an iAUC of 0.981 (95% CI, 0.859–0.967), a PVE of 0.205 (95% CI, 0.151–0.264), and a Brier score of 0.018 (95% CI, 0.008–0.029), with significantly superior risk predictability for thyroid CSS than the TNM-8 staging system, with a C-index of 0.827 (95% CI, 0.622–0.930), an iAUC of 0.879 (95% CI, 0.783–0.962), a PVE of 0.153 (95% CI, 0.095–0.221), and a Brier score of 0.019 (95% CI, 0.009–0.030) (*p* < 0.001).

There are several limitations to this study. First, since this was a retrospective study, it was susceptible to selection bias. Second, Korean patients with PTC have a higher rate of *BRAF* mutations than those from other countries [37,38], and given reports of the synergistic association between *BRAF* V600E and *TERT* promoter mutations [32,33,34], the impact of the *BRAF* mutation alone or its synergistic effect with the *TERT* promoter mutations on CSS may have been reflected in the present study; therefore, it is difficult to apply these results to DTC patients worldwide. Third, since the cohort size and number of cancer-specific deaths per subgroup were small, the statistical power was low; thus, it is necessary to develop this study into a multicenter large-volume study. Nevertheless, it is meaningful that this study is the first to present a new model that combines the mutational profile with the AJCC TNM stage to improve the predictability of CSS with long-term follow-up.

## 5. Conclusions

This is the first work to incorporate the *TERT* mutational status into TNM-8. We propose a new prognostic system, termed TNM-8T, which is superior to the existing system and is easy to apply to real clinics because the contents are not very complicated. This staging system will aid in the precision of management decision making for DTC patients, including decisions on the extent of surgery, prophylactic neck dissection, postoperative radioiodine therapy, and molecular targeted therapy.

## Figures and Tables

**Figure 1 cancers-13-02943-f001:**
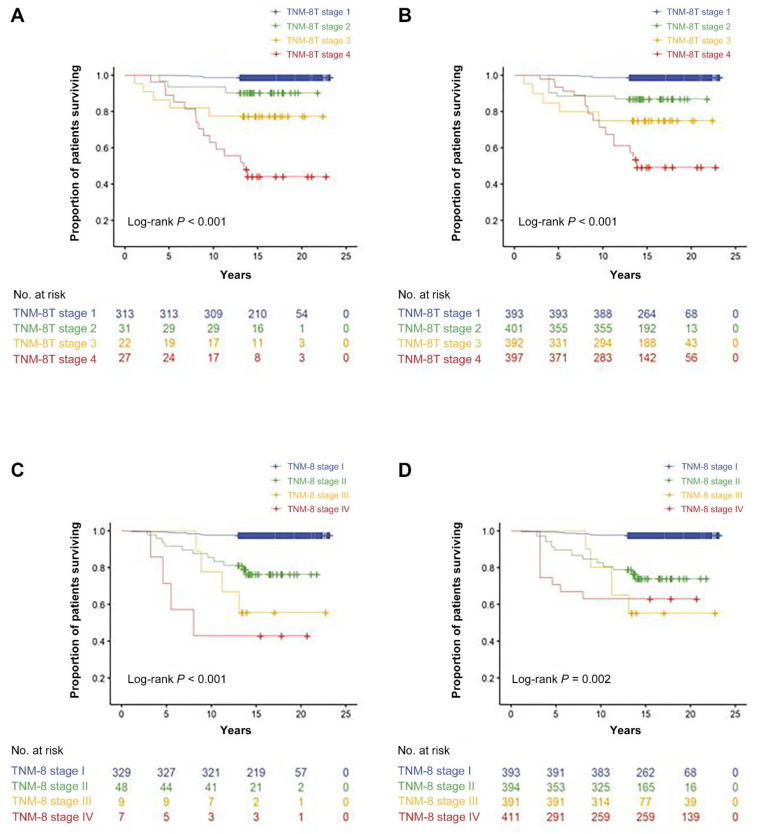
Thyroid cancer-specific survival according to TNM-8T with Kaplan–Meier survival curves (**A**) and adjusted survival curves (**B**). Thyroid cancer-specific survival according to TNM-8 with Kaplan–Meier survival curves (**C**) and adjusted survival curves (**D**).

**Figure 2 cancers-13-02943-f002:**
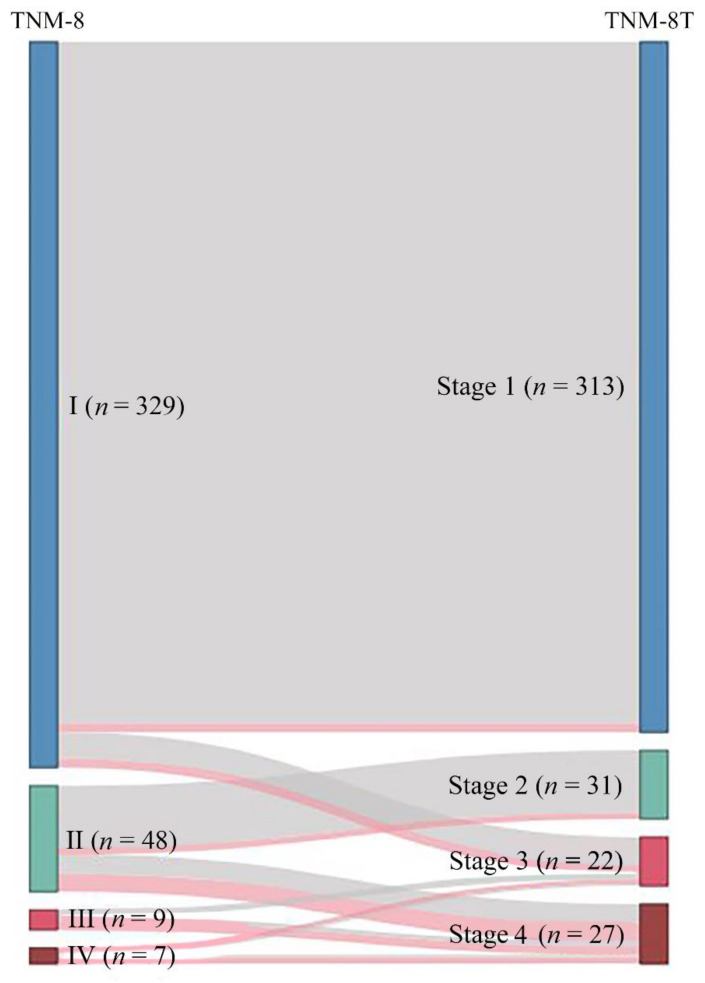
Alluvial flow diagram representing the restaging of patient cohorts from the TNM-8 staging system (Roman numerals) to the TNM-8T staging system (Arabic numerals). Red line indicates deceased patients.

**Table 1 cancers-13-02943-t001:** Baseline characteristics of study patients.

Characteristic	Number (%), (Total = 393)
Age, years	
Median (range)	42.8 (15.8–81.4)
Sex	
Male	329 (83.7)
Female	64 (16.3)
Tumor size, cm	
Median (range)	2.7 (0.4–12.0)
Histological type	
PTC	327 (83.2)
FTC	66 (16.8)
Multifocality	
Absent	286 (72.8)
Present	107 (27.2)
Lymph node metastasis	
Absent	199 (50.6)
Present	193 (49.1)
Missing	1 (0.3)
Extrathyroidal invasion	
Absent	352 (89.6)
Present	41 (10.4)
Distant metastasis	
Absent	370 (94.1)
Present	23 (5.9)
Stage at diagnosis	
I	329 (83.7)
II	48 (12.2)
III/IV	16 (4.1)
*TERT* promoter mutations	
WT	350 (89.1)
Mutant	43 (10.9)
RAI treatment	
Absent	29 (7.4)
Present	364 (92.6)
Incidence related	
Incident cases	27
Persons-year	6366
Incident rate (1000 persons-year)	4
Death	
Survival	366 (93.1)
Death	27 (6.9)

PTC, papillary thyroid cancer; FTC, follicular thyroid cancer; *TERT*, telomerase reverse transcriptase; WT, wild-type; RAI, radioactive iodine.

**Table 2 cancers-13-02943-t002:** Hazard ratios of 6 combinations of TNM-8 stage and *TERT* promoter mutation status for cancer-related death.

Combination	Cancer-Related Death			Unadjusted		Adjusted *	
	Events/Total (%)	10-y CSS	15-y CSS	HR (95% CI)	*p*-Value	HR (95% CI)	*p*-Value
WT/stage I	4/313 (1.3)	98.72	98.72	1.00 (reference)	<0.001	1.00 (reference)	<0.001
WT/stage II	3/31 (9.7)	93.55	90.32	8.05 (1.80–35.98)	0.0063	10.66 (2.37–47.95)	0.002
WT/stage III/IV	1/6 (16.7)	83.33	83.33	15.44 (1.72–138.16)	0.0144	17.82 (2.44–130.18)	0.0045
MUT/stage I	4/16 (25.0)	75.00	75.00	23.66 (5.92–94.63)	<0.001	20.63 (5.17–82.40)	<0.001
MUT/stage II	8/17 (47.1)	70.59	52.29	45.46 (13.68–151.13)	<0.001	55.37 (16.06–190.87)	<0.001
MUT/stage III/IV	7/10 (70.0)	50.00	30.00	78.45 (22.82–269.76)	<0.001	78.37 (22.44–273.67)	<0.001

TNM-8, AJCC tumor-node-metastasis staging system 8th edition; *TERT*, telomerase reverse transcriptase; y, year; CSS, cancer-specific survival; HR, hazard ratio; CI, confidence interval; WT, wild-type; MUT, mutant. * Cox model adjusted for sex and histological type.

**Table 3 cancers-13-02943-t003:** Definition of the TNM-8T groups.

Alternative Grouping	Definition
TNM-8T stage 1	Patients with TNM-8 stage I and wild-type *TERT*
TNM-8T stage 2	Patients with TNM-8 stage II and wild-type *TERT*
TNM-8T stage 3	Patients with TNM-8 stage III/IV and wild-type *TERT* or with TNM-8 stage I and mutant *TERT*
TNM-8T stage 4	Patients with TNM-8 stage II and mutant *TERT* or with TNM-8 stage III/IV and mutant *TERT*

**Table 4 cancers-13-02943-t004:** Hazard ratios of TNM-8T and TNM-8 for cancer-related death.

Alternative Grouping	TNM-8T						
	Cancer-Related Death			Unadjusted		Adjusted *	
	Events/Total (%)	10-y CSS	15-y CSS	HR (95% CI)	*p*-Value	HR (95% CI)	*p*-Value
TNM-8T stage 1	4/313 (1.3)	98.7	98.7	1.00 (reference)	<0.001	1.00 (reference)	<0.001
TNM-8T stage 2	3/31 (9.7)	93.5	90.3	8.05 (1.80–35.97)	0.006	10.62 (2.41–46.71)	0.002
TNM-8T stage 3	5/22 (22.7)	77.3	77.3	21.37 (5.74–79.59)	<0.001	18.57 (5.02–68.70)	<0.001
TNM-8T stage 4	15/27 (55.6)	63.0	44.1	56.40 (18.67–170.36)	<0.001	62.87 (20.66–191.29)	<0.001
**AJCC Stage**	**TNM-8**						
	**Cancer-Related Death**			**Unadjusted**		**Adjusted ***	
	**Events/Total (%)**	**10-y CSS**	**15-y CSS**	**HR (95% CI)**	***p*** **-Value**	**HR (95% CI)**	***p*** **-Value**
TNM-8 stage I	8/329 (2.4)	97.6	97.6	1.00 (reference)	<0.001	1.00 (reference)	<0.001
TNM-8 stage II	11/48 (22.9)	85.4	76.3	10.43 (4.19–25.93)	<0.001	13.20 (5.08–34.30)	<0.001
TNM-8 stage III	4/9 (44.4)	77.8	55.6	20.68 (6.21–68.84)	<0.001	26.99 (8.06–90.42)	<0.001
TNM-8 stage IV	4/7 (57.1)	42.9	42.9	38.24 (11.44–127.75)	<0.001	26.39 (7.84–88.79)	<0.001

TNM-8, AJCC tumor-node-metastasis staging system 8th edition; y, year; CSS, cancer-specific survival; HR, hazard ratio; CI, confidence interval. * Cox model adjusted for sex and histological type.

## Data Availability

The original contributions presented in the study are included in the article/Appendix A. Further inquiries can be directed to the corresponding authors.

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
