# Peer review of "Proposal of a New Prognostic Model for Differentiated Thyroid Cancer with TERT Promoter Mutations"

_cancers, 2021, doi:10.3390/cancers13122943_

Round 1
Reviewer 1 Report
In the present study authors identified a new prognostic stratification system that took into consideration also the presence of the TERT somatic mutation.
There some questions that authors should answer to have the study published
1- 10.9 of cases had a mutation in TERT. Which kind of mutations was found in this series?
2- The indicated prevalence is calculated in the whole series. Did they observed any difference according to the stage of the disease?
3- BRAF mutations have been described in about 50% of PTC cases. Did the authors look for BRAF mutation in their series? It is conceivable that a quite relevant number of cases are BRAF postitive and TERT negative and since it has been largely demonstrated that BRAF mutations play an important role in the progression of the disease the proposed model should take into consideration also the presence of the BRAF mutations. In other words which is the behaviour of BRAF positive/TERT negative cases??
4- Table 4: Adjusted*: please indicate what the asterisk refers to
5- Figure 1, panel A and B: WT1, WT2, WT3&4/MUT1 and MUT2/MUT3&4. Please clearly indicate in the text and in the corresponding legend what they refer to
Author Response
Comments to the Author
In the present study authors identified a new prognostic stratification system that took into consideration also the presence of the TERT somatic mutation.
There some questions that authors should answer to have the study published
- 9 of cases had a mutation in TERT. Which kind of mutations was found in this series?
Answer] Thank for the reviewer’s comments.
Of the total of 43 patients with mutant TERT, 4 patients had C250T and 39 had C228T. We added the above as the following sentence.
[Result section, page 3]:
TERT promoter mutations were identified in 10.9% (43/393, 4 of C250T and 39 of C228T) of patients.
- The indicated prevalence is calculated in the whole series. Did they observe any difference according to the stage of the disease?
Answer] TERT mutations were identified in 4.9% of stage 1, 35.4% of stage 2, and 62.5% of stage 3, showing a statistically significant difference. Following sentence was added in result section.
[Result section, page 3]:
TERT promoter mutations were identified in 10.9% (43/393, 4 of C250T and 39 of C228T) of patients. As the stage increased, the higher the prevalence of TERT mutations was identified (4.9% in stage I, 35.4% in stage II, and 62.5% in stage III/IV). During the median follow-up of 16 years (interquartile range 14-19 years), there were 27 thyroid cancer-related deaths (6.9%).
- BRAF mutations have been described in about 50% of PTC cases. Did the authors look for BRAF mutation in their series? It is conceivable that a quite relevant number of cases are BRAF postitive and TERT negative and since it has been largely demonstrated that BRAF mutations play an important role in the progression of the disease the proposed model should take into consideration also the presence of the BRAF mutations. In other words which is the behaviour of BRAF positive/TERT negative cases??
Answer] The authors do appreciate the reviewer’s thoughtful comments. Our previous paper has an answer to this question (reference #14). In our cohort, the BRAF mutation was confirmed in 75.4% of PTC patients (ref#14, Table 1), and the BRAF positive/TERT negative (BRAF mutation only) group showed a good prognosis for thyroid cancer-specific survival with a 10-year survival rate of 98.9% (ref# 14, figure S2). It also showed a poor prognosis in the TERT mutation only group or TERT and BRAF mutations group. Therefore, in this paper, we focused on TERT mutation and proposed a new prognostic model using it.
- Table 4: Adjusted*: please indicate what the asterisk refers to.
Answer] Thank for the comment. We added the missing description to the footnote of Table 4.
[Table 4, footnote]:
TNM-8, AJCC tumor-node-metastasis staging system 8th edition; y, year; CSS, cancer-specific survival; HR, hazard ratio; CI, confidence interval. * Cox model adjusted for sex and histological type.
- Figure 1, panel A and B: WT1, WT2, WT3&4/MUT1 and MUT2/MUT3&4. Please clearly indicate in the text and in the corresponding legend what they refer to
Answer] Thank you for the helpful comment. Since the definition of alternative grouping in the new prognostic model (TNM-8T) is in Table 3, we changed the labeling in Figure 1 to TNM-8T Stage 1, 2, 3, 4 to make it easier to recognize.
[Figure 1]:
Thank you very much for the helpful comments and suggestions. We did our best to address the reviewer’s requests and hope that it would be the satisfied version for the publication.

Reviewer 2 Report
In this paper the authors present of the use of classification by the TERT promoter mutation for differentiated thyroid cancer. The authors analysed 393 patients who underwent thyroidectomy and with a median follow up of 14 years and found that adding mutation status was more accurate for prediction than the TNM system without this information.
Much of the results are identical to the prior population published in the authors 2021 publication – ref 14, however the analysis is different and of interest and the authors present a patient population that has been treated in a similar fashion. The rationale for the staging system is well described and logical.
The initial regression results of sex and histological type could be included in the supplementary material for completeness.
I find the labelling of the new staging system – TNM-8T 1-4 a little confusing and not easy to differentiate from the AJCC stage, it would be clearer to me if the authors used a clearer label such as Stage 1, Stage 2, Stage 3, Stage 4 and used a consistent labelling system across the text and the figures.
Author Response
Comments to the Author
In this paper the authors present of the use of classification by the TERT promoter mutation for differentiated thyroid cancer. The authors analysed 393 patients who underwent thyroidectomy and with a median follow up of 14 years and found that adding mutation status was more accurate for prediction than the TNM system without this information.
Much of the results are identical to the prior population published in the authors 2021 publication – ref 14, however the analysis is different and of interest and the authors present a patient population that has been treated in a similar fashion. The rationale for the staging system is well described and logical.
- The initial regression results of sex and histological type could be included in the supplementary material for completeness.
Answer] Thank you very much for the thoughtful comment. Since there was an initial regression results in the previous study using the same cohort (ref#14, table 3). We added it to the supplementary material according to the reviewer’s opinion.
[Materials and methods section, page 3, 2.3. Statistical analysis]:
Among the variables with a P value ≤ 0.2 in the univariate analysis, sex and histology were selected as covariates for the multivariate analysis, except for the age, extrathyroidal extension, distant metastasis, and tumor size constituting TNM-8 (Table S1).
Supplemental Table 1. Initial Cox regression analysis of TERT mutation status and clinicopathological variables in patients with DTC.
|
Variables |
Na |
10-year survival rate (%) |
Univariate Cox models |
|
|
HR (95% CI) |
P-value |
|||
|
Sex |
|
|
|
|
|
Female |
329 |
95.1 |
1.00 (reference) |
|
|
Male |
64 |
92.2 |
1.83 (0.78–4.34) |
0.167 |
|
Age, years |
|
|
|
|
|
Per 5 years |
393 |
|
1.75 (1.50–2.06) |
< 0.001 |
|
< 55 |
319 |
97.5 |
1.00 (reference) |
< 0.001 |
|
≥ 55 |
74 |
82.4 |
11.34 (4.96–25.93) |
|
|
TERT mutation |
|
|
|
|
|
Wild-type |
350 |
98.0 |
1.00 (reference) |
< 0.001 |
|
Mutant |
43 |
67.4 |
24.15 (10.56–55.25) |
|
|
BRAF mutation |
|
|
|
|
|
Wild-type |
117 |
92.3 |
1.00 (reference) |
0.331 |
|
Mutant |
199 |
97.0 |
0.64 (0.26–1.57) |
|
|
Histologic type |
|
|
|
|
|
PTC |
327 |
96.6 |
1.00 (reference) |
0.004 |
|
FTC |
66 |
84.8 |
3.15 (1.44–6.88) |
|
|
Multifocality |
|
|
|
|
|
Absent |
286 |
94.4 |
1.00 (reference) |
0.847 |
|
Present |
107 |
95.3 |
0.92 (0.39–2.17) |
|
|
Lymph node metastasis |
|
|
|
|
|
Absent |
199 |
94.0 |
1.00 (reference) |
0.964 |
|
Present |
193 |
95.3 |
1.02 (0.47–2.20) |
|
|
Extrathyroidal extension |
|
|
|
|
|
Absent |
352 |
96.0 |
1.00 (reference) |
< 0.001 |
|
Present |
41 |
82.9 |
4.78 (2.14–10.64) |
|
|
Distant metastasis |
|
|
|
|
|
Absent |
370 |
96.8 |
1.00 (reference) |
< 0.001 |
|
Present |
23 |
60.9 |
10.85 (4.86–24.22) |
|
|
Stage at diagnosisb |
|
|
|
|
|
I |
329 |
97.6 |
1.00 (reference) |
< 0.001 |
|
II |
48 |
85.4 |
10.42 (4.19–25.92) |
< 0.001 |
|
III & IV |
16 |
62.5 |
26.82 (10.03–71.70) |
< 0.001 |
|
Tumor size, cm |
|
|
|
|
|
< 2.0 |
45 |
95.6 |
1.00 (reference) |
0.014 |
|
2.0–4.0 |
293 |
95.9 |
1.24 (0.28–5.38) |
0.776 |
|
> 4.0 |
55 |
87.3 |
3.96 (0.86–18.34) |
0.078 |
Modified form Published information: Table 3 of Cancers. 2020 Feb 5;13(4):648 [14].
TERT, telomerase reverse transcriptase; BRAF, v-Raf murine sarcoma viral oncogene homolog B; DTC, differentiated thyroid cancer; PTC, papillary thyroid cancer; FTC, follicular thyroid cancer; HR, hazard ratio. a The number based on available data for a particular variable in the univariate analysis. b Staging according to the American Joint Committee on Cancer Thyroid Cancer Staging System 8th edition, 2016. HR are referred to the risk of thyroid cancer-specific death
Supplemental Table 2. Hazard ratios of 6 combinations of TNM-8 stage and TERT promoter mutation status for cancer-related death from all pariwise multivariate cox regression
|
|
Adjusted1* |
|
Adjusted2* |
|
Adjusted3* |
|||
|
HR (95% CI) |
P-value |
|
HR (95% CI) |
P-value |
|
HR (95% CI) |
P-value |
|
|
Combinations |
|
<0.001 |
|
|
<0.001 |
|
|
<0.001 |
|
WT/stage I |
1.00 (reference) |
|
|
0.09 (0.02-0.42) |
0.002 |
|
0.06 (0.01-0.41) |
0.005 |
|
WT/stage II |
10.66 (2.37-47.95) |
0.002 |
|
1.00 (reference) |
|
|
0.60 (0.07-4.92) |
0.633 |
|
WT/stage III&IV |
17.82 (2.44-130.18) |
0.005 |
|
1.67 (0.20-13.76) |
0.633 |
|
1.00 (reference) |
|
|
MUT/stage I |
20.63 (5.17-82.40) |
<0.001 |
|
1.94 (0.42-8.95) |
0.398 |
|
1.16 (0.16-8.28) |
0.884 |
|
MUT/stage II |
55.37 (16.06-190.87) |
<0.001 |
|
5.19 (1.35-19.94) |
0.016 |
|
3.11 (0.48-20.25) |
0.236 |
|
MUT/stage III&IV |
78.37 (22.44-273.67) |
<0.001 |
|
7.35 (1.89-28.52) |
0.004 |
|
4.40 (0.64-30.01) |
0.131 |
|
|
Adjusted4* |
|
Adjusted5* |
|
Adjusted6* |
|||
|
HR (95% CI) |
P-value |
|
HR (95% CI) |
P-value |
|
HR (95% CI) |
P-value |
|
|
Combinations |
|
<0.001 |
|
|
<0.001 |
|
|
<0.001 |
|
WT/stage I |
0.05 (0.01-0.19) |
<0.001 |
|
0.02 (0.01-0.06) |
<0.001 |
|
0.01 (0.00-0.04) |
<0.001 |
|
WT/stage II |
0.52 (0.11-2.39) |
0.398 |
|
0.19 (0.05-0.74) |
0.016 |
|
0.14 (0.04-0.53) |
0.004 |
|
WT/stage III&IV |
0.86 (0.12-6.18) |
0.884 |
|
0.32 (0.05-2.10) |
0.236 |
|
0.23 (0.03-1.55) |
0.131 |
|
MUT/stage I |
1.00 (reference) |
|
|
0.37 (0.11-1.28) |
0.116 |
|
0.26 (0.07-0.94) |
0.040 |
|
MUT/stage II |
2.68 (0.78-9.19) |
0.116 |
|
1.00 (reference) |
|
|
0.71 (0.24-2.05) |
0.523 |
|
MUT/stage III&IV |
3.80 (1.06-13.58) |
0.040 |
|
1.42 (0.49-4.11) |
0.523 |
|
1.00 (reference) |
|
TNM-8, AJCC tumor-node-metastasis staging system 8th edition; TERT, telomerase reverse transcriptase; HR, hazard ratio; CI, confidence interval; WT, wild type; MUT, mutant
* A Cox model adjusting for sex, histological type.
Supplemental Table 3. Hazard ratios of TNM-8T for cancer-related death from all pariwise multivariate cox regression
|
|
Adjusted1* |
|
Adjusted2* |
||
|
HR (95% CI) |
P-value |
|
HR (95% CI) |
P-value |
|
|
Alternative grouping |
|
<0.001 |
|
|
<0.001 |
|
TNM-8T stage 1 |
1.00 (reference) |
|
|
0.09 (0.02-0.41) |
0.002 |
|
TNM-8T stage 2 |
10.62 (2.41-46.71) |
0.002 |
|
1.00 (reference) |
|
|
TNM-8T stage 3 |
18.57 (5.02-68.70) |
<0.001 |
|
1.75 (0.40-7.57) |
0.455 |
|
TNM-8T stage 4 |
62.87 (20.66-191.29) |
<0.001 |
|
5.92 (1.74-20.20) |
0.004 |
|
|
Adjusted3* |
|
Adjusted4* |
||
|
HR (95% CI) |
P-value |
|
HR (95% CI) |
P-value |
|
|
Alternative grouping |
|
<0.001 |
|
|
<0.001 |
|
TNM-8T stage 1 |
0.05 (0.01-0.20) |
<0.001 |
|
0.02 (0.01-0.05) |
<0.001 |
|
TNM-8T stage 2 |
0.57 (0.13-2.47) |
0.455 |
|
0.17 (0.05-0.58) |
0.004 |
|
TNM-8T stage 3 |
1.00 (reference) |
|
|
0.30 (0.10-0.84) |
0.022 |
|
TNM-8T stage 4 |
3.39 (1.19-9.65) |
0.022 |
|
1.00 (reference) |
|
HR, hazard ratio; CI, confidence interval
* A Cox model adjusting for sex, histological type.
Supplemental Figure 1. Thyroid cancer-specific survival according to the 6 combinations of TNM-8 stage and TERT promoter mutation status with Kaplan-Meier survival curves (A) and with adjusted survival curves (B).
- I find the labelling of the new staging system – TNM-8T 1-4 a little confusing and not easy to differentiate from the AJCC stage, it would be clearer to me if the authors used a clearer label such as Stage 1, Stage 2, Stage 3, Stage 4 and used a consistent labelling system across the text and the figures.
Answer] The authors do appreciate the reviewer’s valuable comments.
The stage labeling of the new system looks confused with the existing TNM-8, so the TNM-8T is indicated as stage 1,2,3,4 (Arabic numerals), and TNM-8 is indicated I, II, III, and IV (Roman numerals). It was applied consistently throughout the whole paper including tables and figures.
[Result section, page 4]:
Based on the above results, an alternative staging system of four groups termed TNM-8T was generated to predict thyroid CSS: TNM-8T stage 1 (patients with TNM-8 stage I and WT TERT), TNM-8T stage 2 (patients with TNM-8 stage II and WT TERT), TNM-8T stage 3 (patients with TNM-8 stage III/IV and WT TERT or with TNM-8 stage I and mutant TERT), and TNM-8T stage 4 (patients with TNM-8 stage II or III/IV and mutant TERT).
Thank you very much for the helpful comments and suggestions. We did our best to address the reviewer’s requests and hope that it would be the satisfied version for the publication.

Round 2
Reviewer 1 Report
The authors answered to all queries
Author Response
Thanks for your review.